# HIGH-PRECISION REGRESSORS FOR PARTICLE PHYSICS

## ABSTRACT

Monte Carlo simulations of physics processes at particle colliders like the Large Hadron Collider at CERN take up a major fraction of the computational budget. For some simulations, a single data point takes seconds, minutes, or even hours to compute from first principles. Since the necessary number of data points per simulation is on the order of $10^9 - 10^{12}$, machine learning regressors can be used in place of physics simulators to significantly reduce this computational burden. However, this task requires high-precision regressors that can deliver data with relative errors of less than 1% or even 0.1% over the entire domain of the function. In this paper, we develop optimal training strategies and tune various machine learning regressors to satisfy the high-precision requirement. We leverage symmetry arguments from particle physics to optimize the performance of the regressors. Inspired by ResNets, we design a Deep Neural Network with skip connections that outperform fully connected Deep Neural Networks. We find that at lower dimensions, boosted decision trees far outperform neural networks while at higher dimensions neural networks perform significantly better. We show that these regressors can speed up simulations by a factor of $10^3 - 10^6$ over the first-principles computations currently used in Monte Carlo simulations. Additionally, using symmetry arguments derived from particle physics, we reduce the number of regressors necessary for each simulation by an order of magnitude. Our work can significantly reduce the training and storage burden of Monte Carlo simulations at current and future collider experiments.

## 1 INTRODUCTION

Particle physics experiments like those at the Large Hadron Collider at CERN, are running at progressively higher energies and are collecting more data than ever before. As a result, the experimental precision of the measurements they perform is continuously improving. However, to infer what these measurements mean for the interactions between the fundamental constituents of matter, they have to be compared with and interpreted in light of, our current theoretical understanding. This is done by performing first-principles computations for these high energy processes order by order in a power series expansion. After the computation, the resulting function is used in Monte Carlo simulations. The successive terms in the power series expansion, simplistically, become progressively smaller. Schematically, this can be written as:

$$F(\boldsymbol{x}) = f_{00}(\boldsymbol{x}) + \alpha\, f_{01}(\boldsymbol{x}) + \alpha^2\, \{f_{11}(\boldsymbol{x}) + f_{02}(\boldsymbol{x})\} + \dots . \tag{1}$$

where $\alpha \ll 1$ is the small expansion parameter. The term of interest to our current work is the one enclosed by the curly braces in equation (1) which we will refer to as the second-order term[1]. The function, $F(\boldsymbol{x})$, must be evaluated on the order of $10^9 - 10^{12}$ times for each simulation. However, for many processes, evaluating the second-order term, specifically, $f_{02}$, is computationally space- and time-intensive and could take several seconds to compute a single data point. Moreover, these samples cannot be reused leading to an overall high cost of computation for the entire process under consideration. Building surrogate models to speed up Monte Carlo simulations is highly relevant not only in particle physics but in a very large set of problems addressed by all branches of physics using perturbative expansion like the one in equation (1). We give a broader overview of the physics motivation and applications in appendix A.

---

[1] Here *order* refers to the power of the expansion coefficient $\alpha$.

A simple solution to speed up the computation of the functions is to build a regressor using a representative sample. However, to achieve the precision necessary for matching with experimental results, the regressors need to produce very-high accuracy predictions over the entire domain of the function. The requirements that we set for the regressors, and in particular what we mean by *high precision*, are:

**High precision**: prediction error $< 1\%$ over more than $90\%$ of the domain of the function
**Speed**    : prediction time per data point of $< 10^{-4}$ seconds
**Lightweight** : the disk size of the regressors should be a few megabytes at the most for portability

In this work we explore the following novel concepts:

- With simulated data from real physics processes occurring in particle colliders, we study the error distributions over the entire input feature spaces for multi-dimensional distributions when using boosted decision trees (BDT), Deep Neural Networks (DNN) and Deep Neural Networks with skip connections (sk-DNN).

- We study these regressors for 2, 4, and 8 dimensional (D) data making comparisons between the performance of BDTs, DNN and sk-DNNs with the aim of reaching errors smaller than 1% – 0.1% over at least 90% of the input feature space.

- We outline architectural decisions, training strategies and data volume necessary for building these various kinds of high-precision regressors.

In what follows, we will show that we can reduce the compute time of the most compute-intensive part, $f_{11}(\boldsymbol{x}) + f_{02}(\boldsymbol{x})$ (defined in equation (1)), by several orders of magnitude, down from several seconds to sub-milliseconds without compromising the accuracy of prediction. We show that physics-motivated normalization strategies, learning strategies, and invocation of physics symmetries will be necessary to achieve the goal of high precision. In our experiments, the BDTs outperform the DNNs for lower dimensions while the DNNs give comparable (for 4D) or significantly better (for 8D) accuracy at higher dimensions. DNNs with skip connections perform comparably with fully connected DNNs even with much fewer parameters and outperform DNNs of equivalent complexity. Moreover, DNNs and sk-DNNs meet and exceed the high-precision criteria with 8D data while BDTs fail. Our goal will be to make the most lightweight regressor for real-time prediction facilitating the speed-up of the Monte Carlo simulation.

## 2 RELATED WORK

Building models for the regression of amplitudes has been a continued attempt in the particle physics literature in the recent past. boosted decision trees (BDTs) have been the workhorse of particle physics for a long time but mostly for performing classification of tiny signals from dominating backgrounds (Radovic et al., 2018). However, the necessity to use BDTs as a regressor for theoretical estimates of experimental signatures has only been advocated recently (Bishara & Montull, 2019) and has been shown to achieve impressive accuracy for 2D data.

Several other machine learning algorithms have been used for speeding up sample generation for Monte Carlo simulations. Winterhalder et al. (2022) proposed the use of Normalizing Flows (Jimenez Rezende & Mohamed, 2015) with Invertible Neural Networks to implement importance sampling (Müller et al., 2018; Ardizzone et al., 2018). Recently, neural network surrogates have been used to aid Monte Carlo Simulations of collider processes (Danziger et al., 2022). Badger et al. (2022) used Bayesian Neural networks for regression of particle physics amplitudes with a focus on understanding error propagation and estimation. Chen et al. (2021) attempted to reach the high-precision regime with neural networks and achieved 0.7% errors integrated over the entire input feature space. Physics-aware neural networks were studied by Maître & Truong (2021) in an attempt to handle singularities in the regressed functions. In the domain of generative models, GANs (Goodfellow et al., 2014; Springenberg, 2016; Brock et al., 2018) and VAEs (Brock et al., 2018) have been used for sample generation (Butter et al., 2021; Otten et al., 2021).

Similar applications have surfaced in other domains of physics where Monte Carlo simulations are used. Self-learning Monte Carlo methods have been explored by Liu et al. (2017). Applications of

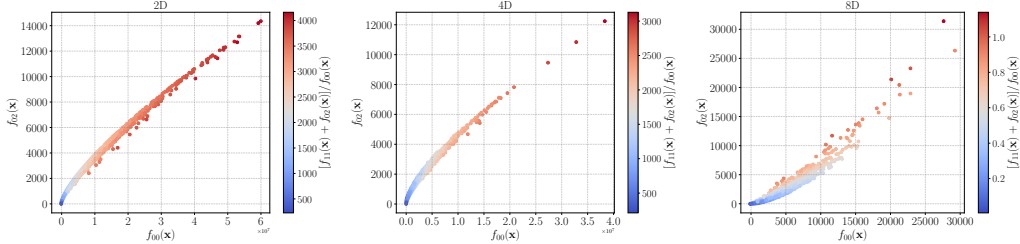

Figure 1: The effects of normalizing second-order term $f_{02}(\boldsymbol{x})$, with the zeroth-order term, $f_{00}(\boldsymbol{x})$. The two functions are highly correlated ($\rho = 0.87$ for 8D, $\rho = 0.91$ for 4D and $\rho = 0.96$ for 2D) and the resulting normalized functions, $f(\boldsymbol{x}) = [f_{11}(\boldsymbol{x}) + f_{02}(\boldsymbol{x})]/f_{00}(\boldsymbol{x})$ are much less peaked.

Boltzmann machines (Huang & Wang, 2017), deep neural networks (Shen et al., 2018) and autoregressive neural networks (Wu et al., 2021) have been seen recently. Stratis et al. (2022) use neural networks in Quantum Monte Carlo simulations to learn eigenvalues of Hamiltonians and the free energy of spin configurations, an application that lies outside the domain of particle physics. However, the primary goal of all these efforts has been to avoid first-principles computation and, hence, reduce compute time while staying below credible error budgets that are set in a problem-specific manner.

In contrast to prior works (Bishara & Montull, 2019; Winterhalder et al., 2022; Butter et al., 2021; Otten et al., 2021), the novelty of our contribution is that we try to attain high precision in the entire domain of the function being regressed with fast and efficient regressors. For that, we compare BDTs and neural networks for functions with 2D, 4D, and 8D input feature spaces. We propose a new architecture which is a DNN with skip connections to avoid the problem of vanishing gradients for deeper neural networks and show that they perform better than fully connected DNNs. We also propose novel methods, derived from physics domain knowledge, for scaling the function being regressed with another function that is computationally inexpensive to calculate and is highly correlated with the function being regressed. We leverage the symmetry properties of the physical process under consideration for the reduction of the number of regressors required to be trained. The applicability of our work goes beyond the domain for which it has been developed and can be used for any application that requires high precision in speeding up simulations or sample generation.

## 3 MODELS: DECISIONS TREES AND NEURAL NETWORKS

In this section, we will develop several methods that will enable us to achieve the high-precision requirements that we set. As a benchmark, we will use the condition: $|\delta| < 1\%$ in over 90% of the domain of the function being regressed[2]. Here, $\delta$ is defined as the difference between the predicted value of the function, $f(\boldsymbol{x})_{\text{predicted}}$, and its true value, $f(\boldsymbol{x})_{true}$, normalized by $f(\boldsymbol{x})_{true}$, or,

$$\delta = \frac{f(\boldsymbol{x})_{\text{predicted}} - f(\boldsymbol{x})_{\text{true}}}{f(\boldsymbol{x})_{\text{true}}} \tag{2}$$

knowing a priori that $f(\boldsymbol{x})_{\text{true}}$ is positive definite. Usually, the performance of a regressor and model comparison in machine learning is done using a single accuracy measure which is a statistical average of the distribution for that accuracy measure over the entire test sample. This, however, does not provide a complete picture of the accuracy of the regressor in high-precision applications. Using error distributions instead of a single number leads to a better criterion for model selection and enhances the interpretability of the model.

**Physics informed normalization:** An attempt to build regressors with the raw data from the Monte Carlo simulations results in a failure to meet the high-precision requirements that we have set. Hence, we have to appeal to a novel normalization method derived from the physics that governs the physical processes. The functions of interest in particle physics processes at colliders are often very highly peaked in one or more dimensions. This makes it quite difficult to build a regressor that will

---

[2]For a more detailed explanation of the precision requirements please read appendix A

retain the desired high precision over the entire domain of the function. This problem cannot be addressed by log scaling or standardizing to zero mean and unit variance since the peaks can be quite narrow and several orders of magnitude greater than the mean value of the function. A regressor trained on the log scaled function provides an error distribution over the entire domain which, when exponentiated, transforms to large errors around the peak. This behavior is not desirable. Normal scaling does not help since the standard deviation of the distribution is much smaller than the peak value, often being several orders of magnitude smaller, making the peak-values outliers.

As a solution, we normalized the second-order contribution with the zeroth-order contribution as defined in equation (1), i.e., we transform to a distribution:

$$f(\boldsymbol{x}) = \frac{f_{11}(\boldsymbol{x}) + f_{02}(\boldsymbol{x})}{f_{00}(\boldsymbol{x})} \, , \tag{3}$$

where $f(\boldsymbol{x})$ is the function that will be regressed. This first-order term, $f_{00}(\boldsymbol{x})$, also has a peak similar to and is highly correlated with the second-order term, $f_{11}(\boldsymbol{x}) + f_{02}(\boldsymbol{x})$, with $\rho \sim 0.9$. Hence, this normalization yields a distribution, $f(\boldsymbol{x})$, that is more tractable to regress. We show examples in figure 1 where one can see that both $f_{00}(\boldsymbol{x})$ and $[f_{11}(\boldsymbol{x}) + f_{02}(\boldsymbol{x})]$ are both very peaked and span several orders of magnitude but their ration spans only one order of magnitude as the two terms are highly correlated. Computation of the first-order term from first principles is numerically inexpensive and does not require regression. Furthermore, we standardize the distribution by removing the mean and scaling to unit variance.

### 3.1 DNN DESIGN DECISIONS

The DNN architectures that we used are fully connected layers with *Leaky ReLU* (Maas, 2013) activation for the hidden layers and linear activations for the output layer. We show a comparative study of activation functions in appendix C where we find that the *Leaky ReLU* outperforms other activation functions like *ReLU* (Nair & Hinton, 2010; Sun et al., 2014), *softplus* and *ELU* (Clevert et al., 2016). We do not consider any learnable activation functions in this work and leave it for a future work. We use the following design decisions:

**Objective function:** we use the mean-square-error loss function without any regularization. While we use linear relative error, $\delta$, to estimate the performance of the model over the entire feature space, our decision to use the mean-square-error loss function is made so as to preferentially penalize outliers and reduce their occurrence.

**Learning rate:** It is necessary to cool down the learning rate as a minimum of the objective function is approached. This is absolutely necessary to search out an optimum that allows for uniformly low error over the entire feature space. For both the DNN and the sk-DNN, we use the Adam optimizing algorithm. Kingma & Ba (2015) discuss an inverse square-root decay rate scaling for the Adam optimizer. We do not find this optimal for this high-precision application. The square-root cooling is quite rigid in its shape as it is invariant to scaling up to a multiplicative constant. Hence, we use an exponential cooling of the learning rate which has an asymptotic behavior similar to the inverse-square-root scaling but its shape is far more tunable. The learning rate is cooled down starting from $10^{-3}$ at the beginning of the training to $10^{-6}$ at 2500 epochs. Much of the learning is completed during the early stages of the training, i.e. within 200 epochs. The $R^2$ score at this point is about 0.5% from the final $R^2$ score ($> 99.9\%$). However, to attain the high-precision requirements, the final stages of the training are necessary and take about $2500 - 3000$ more epochs.

**Training epochs and validation:** An early stopping criterion based on RMSE is used to determine the number of epochs the regressors is trained for with *patience*[3] set to an unusually large number, 200 epochs. We use this large patience to allow the optimizer to possibly move to a better optimum while having a very small learning rate if a better one exists. We first split the data into 20% test set and 80% training and validation set. The latter set is further split into 60% training set and 40% validation set. This results in a 20%-48%-32% split for test, train and validation respectively. The large validation set is necessary to make sure that errors are uniformly low over the entire domain of the function being regressed. For all cases, we use a dataset with 10 million samples.

---

[3]We define patience as the number of epochs after which the training is stopped as no reduction is seen in the RMSE computed from the validation set and the weights and biases are reset to those corresponding to the lowest RMSE.

## 3.2 DNN with skip connections

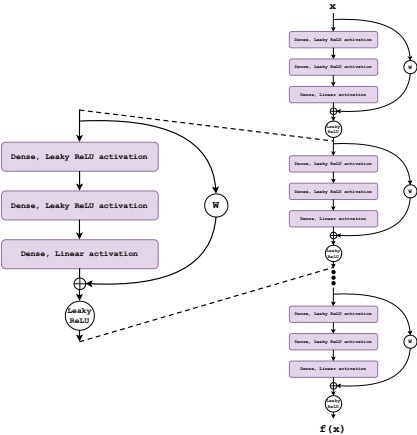

Figure 2: The building block for a DNN with skip connections. The first two layers are fully connected with *Leaky ReLU* activation. The last layer has linear activation and is added to the input through the skip connection before being transformed with a non-linear *Leaky ReLU* function. The matrix $\boldsymbol{W}$ is a weight matrix which is trainable if the input and output dimensions are different for the block and $\boldsymbol{I}$ otherwise. The blocks are stacked sequentially to form the neural network.

In addition to a fully connected DNN, we also experiment with a DNN with skipped connections (sk-DNN) to address the problem of vanishing gradients for deeper neural networks. The building block of the sk-DNN is illustrated in figure 2. Given an input $\boldsymbol{x}$ the output of the block is

$$\boldsymbol{y} = g\left(h(\boldsymbol{x}) + \boldsymbol{W}\boldsymbol{x}\right) \tag{4}$$

where $h(\boldsymbol{x})$ is the output of the third layer with linear activation, $g$ is a non-linear function and $\boldsymbol{W}$ is a trainable weight matrix of dimension $i \times j$ when the input dimension, $i$, is different from the output dimension, $j$, and $\boldsymbol{I}$ otherwise. The structure of this block can be derived from the Highway Network (Srivastava et al., 2015) architecture with the *transform* gate set to $\boldsymbol{I}$ and the *carry* gate set to $\boldsymbol{W}$ for $\dim(\boldsymbol{x}) \neq \dim(\boldsymbol{y})$ and $\boldsymbol{I}$ otherwise. Structurally, the sk-DNN block is similar to a ResNet block (He et al., 2015) with a different set of hidden layers.

We keep the normalization of the target variable and the learning rate decay schedule the same as for the DNN. We also test the sk-DNN with the weight matrix, $\boldsymbol{W}$ fixed with a random initialization of the elements and see no difference in the accuracy of the model and hence keep $\boldsymbol{W}$ trainable.

## 4 Experiments

### 4.1 Physics Simulations

The functions in question are maps, $f_{ij}^{(n)} : \mathbb{R}^n \to \mathbb{R}$, where $n \in \{2, 4, 8\}$ and $i, j \in \{0, 1, 2\}$, cf. equation (1). The domain of the functions, i.e. the feature space, is mapped to the unit hypercube and populated from a uniform distribution. The corresponding datasets are generated using the particle physics simulation code VVAMP (Gehrmann et al., 2015) from first principles using building-block functions that we will refer to as form factors. Apart from the 2D dataset, which is a special case of the 4D one, the same form factors were used to generate the 4D and 8D datasets. The difference between the 4D and 8D feature spaces lies in the physics of the process in question, namely the number of external particles the functions describe. The regressor of the 4D functions, $g_{ij}^{(4)} \approx f_{ij}^{(4)}$, can be used to generate the 8D functions, $f_{ij}^{(8)}$, after multiplying by two other (exact) functions that are computationally inexpensive to calculate and summing them.

The number of resulting functions, technically called helicity amplitudes, depends on the dimension as shown in table 1. While the number of required regressors for the 4D feature space is the largest, it also offers the most flexibility for downstream physics analyses. To generate the 8D functions,

| Symmetry properties reduce the number of required functions | | | |
|---|---|---|---|
| Dimensionality | Total functions | Independent functions | Sum is physical? |
| 2D | 18 | 5 | Yes |
| 4D | 162 | 25 | No |
| 8D | 8 | 4 | Yes |

Table 1: The number of total and independent functions that arise at 2, 4, and 8D and the number of independent functions, a minimal subset from which the other functions can be generated by re-mapping the feature space.

more details of the process have to be specified during data generation which is then frozen into the regressor. Consequently, different physics analyses will require different regressors. By contrast, the 4D regressors are more general-purpose and do not contain *any* frozen physics parameters.

The smaller number of necessary functions in the third column of table 1 is obtained by leveraging the symmetry properties discussed below derived from physics domain knowledge. For the data used in this analysis, it stems from the symmetries manifest in the scattering process that was simulated. The last column indicates whether summing the functions has a physics meaning; in the cases where it does, i.e. 2D and 8D, only the single regressor of the sum of the functions is required.

**Symmetry properties:** the full set of functions, $f_{ij}^{(n)}$, for any dimension, $n$, is over complete. Pairs of functions can be mapped into one another via particular permutations of the external particles the process describes. This translates into a linear transformation on the second coordinate, $x_2$, independently and in combination with the permutation of the third and fourth coordinates, $x_3$ and $x_4$, in feature space. For example, in 4D, two permutations $\pi_{12} : p_1 \leftrightarrow p_2$ and $\pi_{34} := p_3 \leftrightarrow p_4$, where $p_i$ is a particle with label $i$ reduces the number of independent functions from 162 to 25.

| Permutation | particle symmetry | coordinate symmetry |
|---|---|---|
| $\pi_{12}$ | $p_1 \leftrightarrow p_2$ | $x_2 \to 1 - x_2$ |
| $\pi_{34}$ | $p_3 \leftrightarrow p_4$ | $x_2 \to 1 - x_2$ and $x_3 \leftrightarrow x_4$ |

**Computational burden of Monte Carlo simulations.** Generating the 2D, 4D and 8D datasets required 144 hours on 96 `AMD EPYC 7402` cores for 13 million data points per set. This had to be done twice, once for the 2D dataset and once for the 4D and 8D datasets which were generated from the same computationally intensive form factors which have to be calculated from first principles. In contrast, the regressors that we build generate a million samples in a few seconds to a few minutes on any desktop computer.

## 4.2 BDT vs. DNN vs. sk-DNN

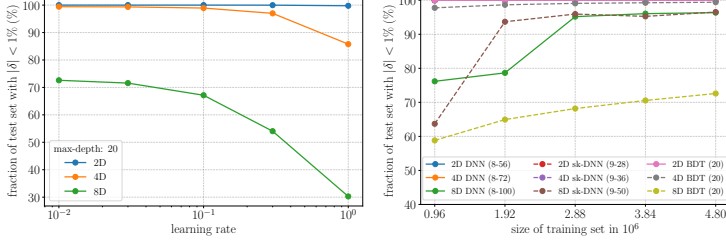

Figure 3: **Left panel:** the effect of learning rate on achieving high precision with BDTs. The learning rate is not an important parameter for low dimensions but is significant for higher dimensions. **Right panel:** the data volume required to train the different regressors. For lower dimensions small volumes of data ($< 1M$) is sufficient. However, for higher dimensions, a lot more data is necessary.

We use `XGBoost` (Chen & Guestrin, 2016) to implement the BDTs. In varying the architecture of the regressors, we focus on the *max-depth* of the BDT which is a hyperparameter that controls the

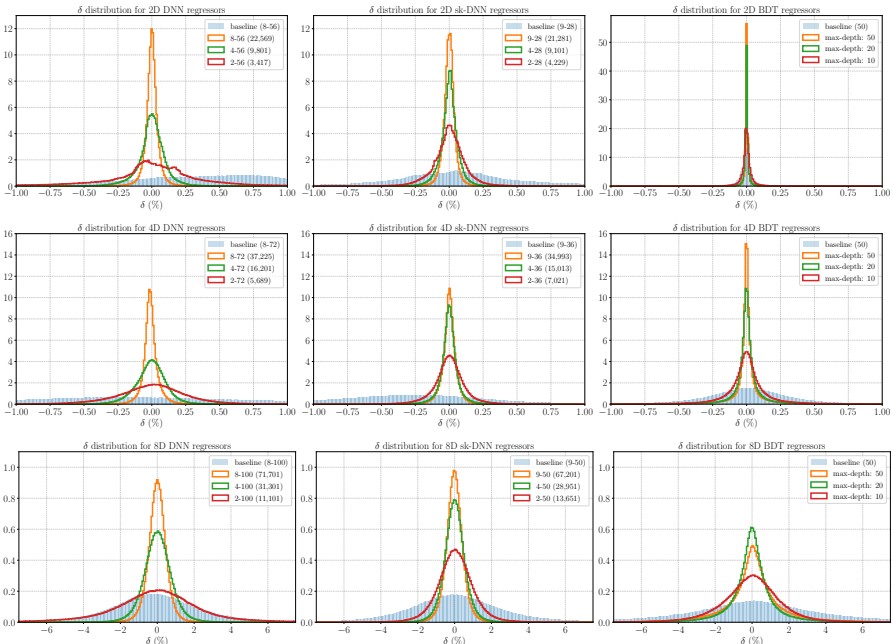

Figure 4: The $\delta = (f(\boldsymbol{x})_{\text{predicted}} - f(\boldsymbol{x})_{\text{true}})/f(\boldsymbol{x})_{\text{true}}$ distribution for the various 2D (**upper panels**), 4D (**middle panels**), and 8D (**lower panels**) regressors. The labels for the DNN and sk-DNN designate *depth-width (number of parameters)* where depth is the number of layers for the DNN and the number of blocks for the sk-DNN. For the BDTs, the labels denote *max_depth: N* with max_dept being the maximum depth of the trees. Detailed analyses can be found in the text.

maximum depth to which a tree grows in a boosted ensemble. If figure 3 we show how changing the learning rate and the training data volume changes the accuracy of the trained BDT models. In the final version of our experiments, we use a learning rate of 0.01 and 10 million data points of which 48% is used for training, 32% is used for validation and early stopping and 20% is used for testing. More details on hyperparameter correlation and selection can be found in appendix B.

For the neural networks, we focus on the depth, width and number of trainable parameters in the regressor (denoted as width-depth (trainable parameters) in the tables and figures). The depth of the sk-DNN denotes the number of sequential sk-DNN blocks in the regressor and not the total number of layers. The width of the sk-DNNs is chosen to be half the width of the DNNs and the depth of the sk-DNN is adjusted so that they have approximately the same number of parameters as the DNNs with similar depth. An sk-DNN with 2 blocks is an exception and has more parameters than the corresponding DNN with 2 layers. The data strategy remains the same as for the BDTs.

To compare the performance of the regressor we use the distribution of $\delta$ (defined in equation (2)). We focus on this distribution as it is important for the high-precision requirement to identify the fraction of test data that has large errors. We will identify the following statistics:

$|\delta| < 1\%$: the fraction of the test set that has $\delta$ less than 1%

$\mu_\delta$: the mean of the $\delta$ distribution

$\sigma_\delta$: the standard deviation of the $\delta$ distribution

**Baselines:** To understand the efficacy of the optimization strategies that we developed, we build a baseline without any optimization for BDTs, DNNs and sk-DNNs. We do not normalize the data as described in section 3, rather, we only log scale the data. We set the train-validation split to 80%-20%. For the BDTs, we use an ensemble with max-depth = 50, set the learning rate to 0.1. For the DNNs and sk-DNNs, we fix the learning rate of the Adam optimizer at $10^{-3}$, lower the patience to 10 rounds, and use the most effective architecture chosen from amongst the high-precision regressors. The results are presented in table 2 and figure 4. We see that without the optimizations the regressors perform very poorly.

| | **2D** | $|\delta| < 1\%(\%)$ | $|\delta| < 0.1\%(\%)$ | $\mu_\delta(\%)$ | $\sigma_\delta(\%)$ |
|---|---|---|---|---|---|
| **DNN** | 2-56 (3,417) | 95.54 | 32.43 | 0.0114 | 0.52 |
| | 4-56 (9,801) | 99.95 | 75.58 | $-0.0005$ | 0.13 |
| | 8-56 (22,569) | 99.99 | 94.43 | 0.0 | 0.06 |
| | baseline (8-56) | 77.16 | 9.01 | 0.1361 | 1.83 |
| **sk-DNN** | 2-28 (4,229) | 99.92 | 67.71 | 0.0001 | 0.14 |
| | 4-28 (9,101) | 99.99 | 87.95 | 0.0012 | 0.07 |
| | 9-28 (21,281) | 100.0 | 95.72 | $-0.0005$ | 0.05 |
| | baseline (9-28) | 90.79 | 15.31 | 0.0173 | 1.39 |
| **BDT** | max-depth: 10 | 100.0 | 95.01 | $-0.0001$ | 0.05 |
| | max-depth: 20 | 100.0 | 99.1 | 0.0 | 0.03 |
| | **max-depth: 50** | 100.0 | 99.16 | 0.0 | 0.02 |
| | baseline (50) | 99.91 | 94.04 | $-0.0045$ | 0.1 |
| | **4D** | | | | |
| **DNN** | 2-72 (5,689) | 99.18 | 34.34 | 0.002 | 0.32 |
| | 4-72 (16,201) | 99.97 | 66.42 | $-0.0068$ | 0.13 |
| | 8-72 (37,225) | 100.0 | 91.58 | $-0.0096$ | 0.07 |
| | baseline (8-72) | 88.67 | 13.63 | 0.0449 | 1.1 |
| **sk-DNN** | 2-36 (7,021) | 99.96 | 69.23 | 0.0004 | 0.13 |
| | 4-36 (15,013) | 100.0 | 89.24 | $-0.0009$ | 0.07 |
| | **9-36 (34,993)** | 100.0 | 92.85 | 0.0007 | 0.06 |
| | baseline(9-36) | 84.99 | 10.81 | 0.2701 | 1.11 |
| **BDT** | max-depth: 10 | 99.26 | 66.16 | 0.0014 | 0.22 |
| | max-depth: 20 | 99.41 | 81.34 | 0.0016 | 0.18 |
| | max-depth: 50 | 99.4 | 83.19 | 0.0017 | 0.18 |
| | baseline (50) | 95.85 | 27.8 | 0.0023 | 0.55 |
| | **8D** | | | | |
| **DNN** | 2-100 (11,101) | 37.2 | 3.95 | 0.1322 | 4.13 |
| | 4-100 (31,301) | 82.37 | 11.64 | 0.029 | 1.05 |
| | 8-100 (71,701) | 94.22 | 18.12 | 0.0016 | 0.6 |
| | baseline (8-100) | 31.97 | 3.3 | 0.799 | 4.38 |
| **sk-DNN** | 2-50 (13,651) | 72.76 | 9.31 | 0.049 | 1.5 |
| | 4-50 (28,951) | 90.98 | 15.69 | 0.0035 | 0.7 |
| | **9-50 (67,201)** | 94.94 | 19.36 | $-0.0063$ | 0.56 |
| | baseline (9-50) | 30.91 | 3.23 | $-0.547$ | 4.85 |
| **BDT** | max-depth: 10 | 51.68 | 5.95 | 0.0921 | 2.91 |
| | max-depth: 20 | 72.6 | 12.06 | 0.0577 | 1.85 |
| | max-depth: 50 | 62.15 | 9.61 | 0.1505 | 2.36 |
| | baseline (50) | 22.33 | 2.3 | 1.3953 | 13.35 |

Table 2: Parameters extracted from the error distributions shown in figure 4. Predictions from 1 million test samples were used to generate these statistics. The best-performing models for each of 2D, 4D, and 8D are marked in bold. More details regarding the notations are available in the text.

**Key results:** we present the results of the experiments in table 2 and figure 4. We show the distribution of errors over two variables, square root of the center-of-mass energy, $\sqrt{s} \mapsto x_1$, and $\cos\theta \mapsto x_2$ in figure 5. It is clear that the BDTs far outperform the DNNs in 2D. However, at 4D and 8D the sk-DNN not only outperforms the fully connected DNNs, but also outperforms the BDTs as can be seen from the distributions in figure 4 and the numbers in table 2. While at 4D the improvement of accuracy from the DNN and sk-DNN is marginal over the BDTs, at 8D the improvement of accuracy is quite significant. One major disadvantage of the BDTs is that they take up significant disk space as the ensemble grows large, especially at higher dimensions, which is necessary for high-precision applications but affects their portability. Hence the sk-DNNs are a good solution for having a portable, yet accurate regressor that meets the thresholds we set at the beginning of the work.

## 5   CONCLUSION

With Monte Carlo simulation in Physics being time and resource intensive, a distinct necessity of building regressors for speeding up the simulations exists. We carefully examine the requirements of high precision for these regressors and lay down design strategies to achieve the necessary benchmarks. We use domain knowledge from particle physics to determine normalization strategies, apply symmetry arguments to reduce the number of necessary regressors, and set benchmarks for high-precision regression.

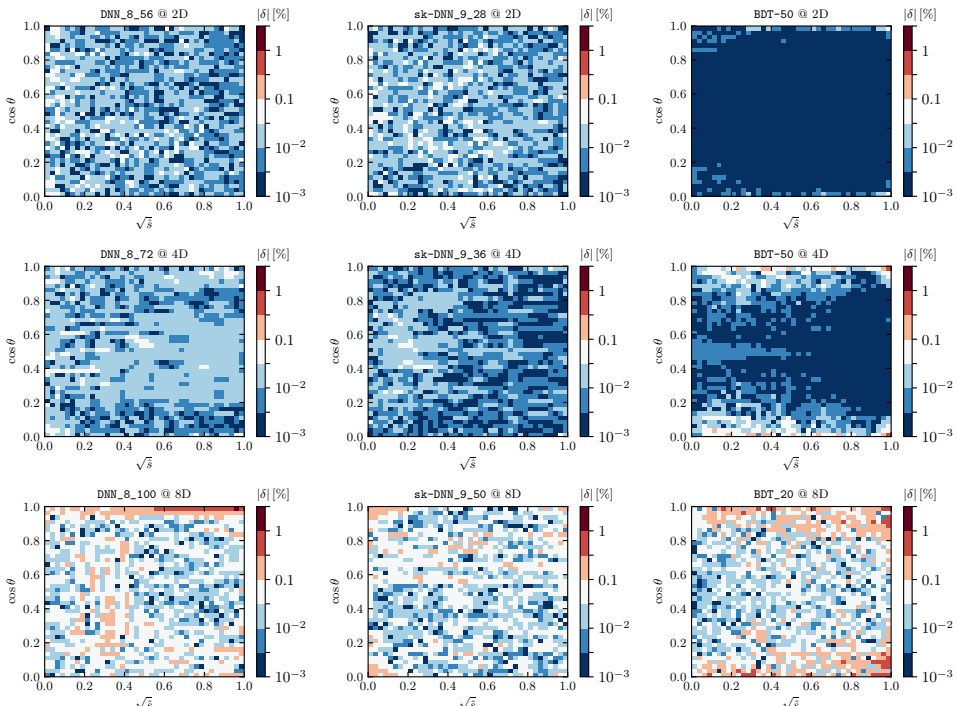

Figure 5: The $\delta = (f(\boldsymbol{x})_{\text{predicted}} - f(\boldsymbol{x})_{\text{true}})/f(\boldsymbol{x})_{\text{true}}$ distribution for the various 2D (upper panels), 4D (middle panels) and 8D (lower panels) regressors. The errors are averaged over each bin.

We show that boosted decision trees are reliable workhorses that can easily outperform DNNs at lower dimensions even when very large and complex neural networks are used. However, this edge that BDTs have over neural networks tends to fade at higher dimensions especially when DNNs with skip connections are used. In fact, for 4D and 8D data, sk-DNNs outperform both DNNs and BDTs and exceed the benchmark of $\delta < 1\%$ over 90% of the domain of the function. Moreover, sk-DNNs are capable of outperforming DNNs of higher complexity as can be seen from table 2.

The primary disadvantage of BDTs is that for higher dimensions the ensemble of trees grows large enough to take a significant amount of disk space, often $> 1$ GB, affecting the portability of the regressor if it is intended to be used as part of a Monte Carlo simulation package. On the other hand, the disk space occupied by a neural network stay below a few megabytes, making them a lot more portable. In summary, the important conclusions of our work are:

- High precision regressors required to speed up Monte Carlo simulations by factors of $10^3$ – $10^6$ are better optimized by leveraging physics domain knowledge and symmetry arguments.
- BDTs outperform DNNs at lower dimensions but start to make large errors in predictions in parts of the function domain at higher dimensions. While fully connected DNNs perform relatively well at higher dimensions, a DNN with skip connections outperforms both BDTs and fully connected DNNs at 4D and 8D.
- sk-DNNs can outperform DNNs with a larger number of parameters.
- Compared to the few seconds that it takes for a single sample generation during a Monte Carlo simulation, the regressors we design can provide precise predictions in milliseconds to microseconds.

In this work, we aimed at reaching the desired precision but by no means have we exhausted the possibilities of achieving even higher precision. As future directions, surveying a wider gamut of activation functions, the modifications of the likelihood with possible physics constraints or symmetry arguments or further reducing the number of models by simultaneously predicting a set of functions from a single neural network might be directions that can be explored in detail.

CONTRIBUTION TO SUSTAINABILITY

Monte Carlo simulations of physics processes leave a very large carbon footprint. It is estimated that about 50% of the energy budget of each experiment at the Large Hadron Collider is consumed by such simulations. Hence, our work directly contributes to reducing the carbon footprint significantly through a much more efficient way of generating these events.

Generating the 2D, 4D and 8D datasets required 144 hours on 96 `AMD EPYC 7402` cores for 13 million data points per set. This had to be done twice, once for the 2D dataset and once for the 4D and 8D datasets which were generated from the same first principles computation. In contrast, the regressors that we build generate a million samples in a few seconds to tens of seconds on any desktop computer. The regressors we build can be trained on personal computers with a few CPU threads and a single GPU in about a day as our focus has been to build lightweight models. No special hardware is required to train or test these regressors. Given that these Monte Carlo simulations have to be done thousands of times during the life cycle of a single analysis, the regressors can significantly reduce the carbon footprint from energy consumption without any significant compromise to the precision necessary for quantitative scientific research.

REPRODUCIBILITY STATEMENT

The code and the data will be made publicly available after the review process to maintain the double-blind requirements.

AUTHOR CONTRIBUTIONS

ACKNOWLEDGMENTS

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

## Appendix

## A THE PHYSICS BACKGROUND

The goal of this appendix is to provide the context in which the regressors discussed in this paper are needed and thus elucidate the requirements laid out in the introduction. Namely,

1. that the prediction error relative to the exact value be $< 1\%$,
2. that this be the case for 90% of the domain of the function,
3. that the evaluation time be faster than $\mathcal{O}(10^{-3})$ seconds,
4. and, finally, that the disk size of the regressors be on the order of megabytes rather than gigabytes.

These regressors will be used as surrogate models for exact functions that are numerically slow to evaluate. As a result of their (extreme) slowness, these exact functions, which are used in Monte Carlo simulations, are by far the biggest bottleneck in the simulation.

**The physics context.** The theoretical model that describes the fundamental particles and their interactions is called the Standard Model of particle physics. This model is an example of a quantum field theory; what this means exactly is not crucial here. Rather, the important feature of this theory is that computing observables (i.e., outcomes of experiments) cannot, in general, be done *exactly* because such calculations are not tractable for several reasons the explanation of which goes beyond the scope of this work. The usual way of obtaining predictions is by expanding the theory as a power series in a small expansion parameter and computing higher orders in this expansion to improve the accuracy of the prediction. Such perturbative expansions are ubiquitous in physics in general since only a few systems, most notably, e.g., the simple harmonic oscillator and the two-body inverse $r^2$ problem can be solved exactly. A very large fraction of physics problems spanning atomic physics, nuclear physics, condensed matter physics, astrophysics, cosmology, hydrodynamics, electrodynamics, quantum mechanics, complex systems etc. requires the use of perturbative expansions where the higher order terms are very tedious and slow to compute. Hence, the methods

we develop here are more broadly applicable in any problem where a perturbative expansion is used and/or a function that requires a very large number of evaluations is very slow to evaluate and a certain threshold of precision is required.

The slow functions that are the focus of this work arise at second order in this power series expansion. The number of terms produced at each order rapidly increases and the complexity of the mathematical functions that appear also increases. For example, at first order in the expansion (if the zeroth order is a so-called tree process), polylogarithms of at most order 2 can appear. At second order, higher order polylogarithms appear. On top of the fact that these functions are time-consuming to evaluate numerically, large cancellations between these functions typically exist which requires using arbitrary precision arithmetic libraries to circumvent the infamous 'catastrophic cancellation' problem in numerical analysis.

For example, the time penalty for improving the accuracy of the prediction of the rate of production of four electrons by including the second-order term is a factor of 1500! For details, please see table 11 in the journal version of Ref. Grazzini et al. (2018). So here lies the logic behind requirement 3: the second-order functions typically take $1 - 10$ seconds per point to evaluate while all other functions in the Monte Carlo simulation typically take milli-seconds. Therefore, **the bottleneck is removed** if the surrogate model takes $10^{-3}$ second per point or less to evaluate.

**The accuracy requirement.** There are many sources of uncertainty that propagate to the final prediction. Roughly speaking, there are systematic errors of order 1% that cannot be reduced at the moment and for the foreseeable future. There are also statistical errors inherent in the finite samples produced by Monte Carlo simulations. The goal is to strive to have Monte Carlo statistical errors much smaller than 1%, say 0.1%. Since the contribution of the second order functions is of order 10%, then it is **sufficient for the surrogate models to be accurate to 1%** in order for the error on the total prediction (including the zeroth and first order) to be of order 0.1%. While this precision would be good to have in the entire domain of the function it is not necessary given the error margins we aim for. Assuming that the errors in the predictions made by the model follow a Gaussian distribution, we can safely set the threshold to 1% error over 90% of the function domain. We checked, after the fact, that the Gaussian assumption is approximately realized (cf. figure 4). An elaboration of this requires a discussion of specific integrals for specific scattering processes in particle physics that we shall leave for a more particle-physics-oriented work.

**Portability.** In contrast with the speed and accuracy requirements (items 1, 2, & 3), the requirement that the disk size of the models be of order megabytes (item 4) is desirable but not a strict requirement. In practice, to implement the surrogate models discussed in this work into Monte Carlo codes, several regressors are required. Since these codes must be downloaded locally by the users, it is desirable that the disk size remain small. As shown before BDTs can reach several GB in compressed model format for higher dimensional data. Hence, in this work, we focus on building specific neural networks that are a lot more portable.

## B  HYPERPARAMETER SURVEYS FOR BOOSTED DECISION TREES

**Maximum depth of trees in the ensemble:** The BDT models are trained with an early stopping condition which stops the growth of the trees once the RMSE stops decreasing after checking for its decrease for 25 rounds. This makes the hyperparameters used to train a BDT correlated to a certain extent. For example, a decrease in the learning rate increases the number of trees grown till the optimum is reached. This can be seen from figure 6. However, as one increases the maximum depth to which each tree can grow the number of total trees grown decreases. The number of nodes of a tree grows exponentially with the depth of the trees and, hence, allowing for a larger maximum depth of the trees results in a much larger disk size for the trained models. This is aggravated further with higher dimensional data. Therefore, when portability is a concern, BDTs cannot be used for high-precision applications for higher dimensional data.

**Learning rate and maximum depth of trees:** When exploring the learning rate for the BDT models in figure 6, we find that, initially, with decreasing learning rate, starting at 1, the accuracies of the trained models increase but after a point, the accuracy decreases. This is evident for shallower trees. We also note that the accuracies of the models increase with the maximum depth of the trees but

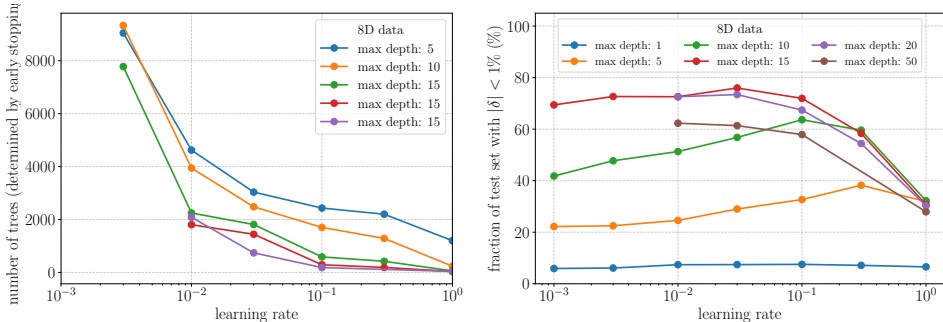

Figure 6: **Left panel:** The variation of the number of trees grown in a BDT ensemble increases rapidly with decreasing learning rate when using an early-stopping criterion. Note: The learning rate for BDTs with maximum depths 20 and 50 could not be reduced below 0.1 as the disk size of the memory consumption while training the models with 8D data gets too large for a single node in a high-performance computing cluster. **Right panel:** Larger maximum depth for BDT ensembles gives better accuracy up until a certain value and then the accuracy falls. The optimal value for maximum depth seems to be 15 or 20. Also, the learning rate has an optimal after which it decreases or plateaus.

only up to a certain depth. In the example in the right panel of figure 6 we use the 8D data and we see that the accuracy of the model increase till a maximum depth of 15 and then decreases.

## C  DEPENDENCE ON ACTIVATION FUNCTIONS

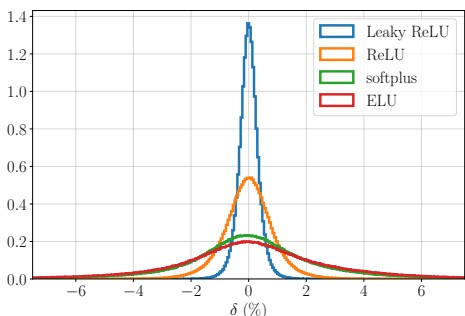

Figure 7: A comparison between *Leaky ReLU*, *ReLU*, *softplus* and *ELU* for the 8D data using an sk-DNN with 9 blocks of width 50. The *Leaky ReLU* activation function outperforms any other activation function and we use it for all experiments with DNNs and sk-DNNs in our work

We performed tests for various activation functions keeping all other hyperparameters and data strategies the same. We use the 8D dataset with a 9-deep and 50-wide sk-DNN on an exponential learning rate schedule and data normalized using equation (3). We explore only non-trainable activation functions like the *ReLU*, *Leaky ReLU*, *ELU* and softmax activations functions. The last three were chosen as they are similar to *ReLU* and have shown improved learning abilities in several domains Maas (2013); Sun et al. (2014); Clevert et al. (2016). As in the other experiments, the models were trained with an early-stopping criterion. From figure 7 we see that the *Leaky ReLU* activation function far outperforms all other activation function with a narrower error distribution. Hence, for all experiments in this work we use the *Leaky ReLU* activation function.

