# OpenReview forum: "High-Precision Regressors for Particle Physics"
_ICLR.cc/2023/Conference — Submitted to ICLR 2023_

### Official Review · Reviewer_XmXm · 2022-10-24

**Confidence:** 5
**Correctness:** 3
**Technical Novelty And Significance:** 2
**Empirical Novelty And Significance:** 2
**Recommendation:** 1

**Clarity, Quality, Novelty And Reproducibility:**

The clarity of the introduction could be improved. The work is original, but not particularly relevant to ICLR.

**Strength And Weaknesses:**

Strengths:
- The authors use a large dataset generated from high-energy physics. These datasets can be quite useful in comparing ML methods.

Weaknesses:
- The introduction could be written more clearly. I believe the task is to train a surrugate model that can be used to approximate some function that is traditionally computed using Monte Carlo methods, but the description of the problem and the related work obscure this.
- This is an example of a regression problem. The authors use this particular application as a case study in the ability to perform regression with neural networks, but ultimately any general claims will need much more supporting evidence (or theory). I don't think this application alone will be of interest to the community.
- The intput-normaliation strategy certainly seems useful, but is particular to this data, and not a general technique that is of interest to the community.
- The other modifications including skip connections are standard architectural design choices that are frequently optimized through hyperparameter tuning. Thus, the fact that they lead to improved performance is an interesting data point, but not enough to be of high interest in the community.

**Summary Of The Paper:**

This paper develops a deep neural network model for a regression problem in high-energy physics. The paper highlights the particular constraints of the application which include high precision, speed, and memory efficiency. The method is compared on dataset from physics against standard neural networks and XGBoost.

**Summary Of The Review:**

While this seems like a nice application of machine learning, there is not enough here to justify publication at ICLR.

---

> ### Author Response · Authors · 2022-11-16
> **Response to reviewer 4**
>
> We thank the reviewer for taking a careful look at our work. We would like to address some of the concerns the reviewer has raised. We do so below.
> _____________
> - “The authors use a large dataset generated from high-energy physics. These datasets can be quite useful in comparing ML methods.”
>
> We thank you for pointing this out. Indeed we are releasing all the data for public use and these are real-world datasets generated by simulators that are the gold standard in particle physics simulations.
>
> _____________
>
> - "The introduction could be written more clearly. I believe the task is to train a surrugate model that can be used to approximate some function that is traditionally computed using Monte Carlo methods, but the description of the problem and the related work obscure this."
>
> We realized that the explanation in the introduction is not sufficient and we are adding an appendix to clarify the physics case and the utility of our work
>
> This is an example of a regression problem. The authors use this particular application as a case study in the ability to perform regression with neural networks, but ultimately any general claims will need much more supporting evidence (or theory). I don't think this application alone will be of interest to the community.
>
> We do agree with the reviewer that this is an empirical work without the development of a theory behind it. As it is known, the underlying theory behind universal approximation is not well understood and some progress has been made in recent years. For example, this very recent JMLR paper: https://www.jmlr.org/papers/volume23/21-1404/21-1404.pdf tries to study exactly this problem from a theoretical standpoint. However, the thresholds they set for the networks are very high and will result in much more complex networks that we have with orders of magnitude more parameters. The goal of our work, as you correctly pointed out, is to make efficient regressors that are precise enough and we believe the best path to this is through empirical studies.
>
> ___________
> - "The intput-normaliation strategy certainly seems useful, but is particular to this data, and not a general technique that is of interest to the community."
>
> Yes, we exploit the correlation between functions to normalize a function that is highly peaked and intractable for building high-precision regressors. We understand that this will not be the case for any general dataset. However, this is often the case for the domain of problems we aim to address, namely, amplitudes for the scattering of particles at colliders. Moreover, we point out that in the case that such correlations exist in a perturbative expansion (in the domain of physics in general or in any other data), such normalization can be very useful. To our knowledge, the idea of normalizing a function with a correlated one that is much cheaper to compute to increase the precision of a regressor has not been explored before in the context of training ML models.
>
> __________
> - "The other modifications including skip connections are standard architectural design choices that are frequently optimized through hyperparameter tuning. Thus, the fact that they lead to improved performance is an interesting data point, but not enough to be of high interest in the community."
>
> As we mention in the text. Skip connections have been around since the paper  “Highway Networks” was published in early 2015. Residual networks came in late 2015 and were a special application of the same concept to image data. We do not propose a new method but we show its applicability in a domain and architecture it has not been used before. To our knowledge, skip connections have not been used with simple DNN using dense layers only. While this is an incremental improvement in methodology, we do believe it brings with it some degree of novelty.
>
> _________
>
> - “The clarity of the introduction could be improved. The work is original, but not particularly relevant to ICLR.”
>
> We are adding an appendix to clarify the motivations and will augment the information that we provide in the introduction. The reason we believed this would be fit for ICLR is that it explores a problem that is core to fundamental sciences (MC simulators are quite popular for several very complex problems and speeding them up with regressors is an urgent problem to address) and has not been sufficiently explored judging by the lack of ML literature on building architectures dedicated to this. In a way, this is an empirical exploration of how to approach the universal approximation limit without making the models extremely complex.
>
> __________
>
> "Correctness: 3: Some of the paper’s claims have minor issues. A few statements are not well-supported or require small changes to be made correct."
>
> We would like to know what these not-well-supported statements are. While we have been careful with our claims, it is entirely possible we might have erred.

---

### Official Review · Reviewer_XeYb · 2022-10-24

**Confidence:** 3
**Correctness:** 4
**Technical Novelty And Significance:** 3
**Empirical Novelty And Significance:** 2
**Recommendation:** 6

**Clarity, Quality, Novelty And Reproducibility:**

Clarity: good

Quality: fair

Originality: good in ML for particle physics, may not be novel in general ML

Reproducibility: fair

**Strength And Weaknesses:**

Strengths:

The paper addresses an important problem in particle physics. The physics-informed design of the architecture and data normalization introduces useful inductive bias and helps improving the performance. The speedup is significant which is important in the field.

Weaknesses:

In terms of novelty, w.r.t. the field of machine learning for particle physics, this method may be novel and significant (I'm not an expert in particle physics). On the other hand, in terms of machine learning, the novelty is limited, since the residual connection is a commonly used architecture. The physics-informed normalization is interesting, but may be limited in its generality to more general problems. I'm OK with judging the novelty either way.

The limited novelty may be compensated by a more comprehensive empirical experiments. The authors are encouraged to explore more hyperparameters and compare with more baselines. For example, the authors may explore how activation function affect the performance, since activation is extremely important in regression tasks. In my experience, for regression tasks, ReLU does not necessarily work well since it essentially models a piecewise linear function. The activation of leakyReLU, SiLU [1], ELU [2], Rational activation [3] are typically very good candidate activation for regression tasks, especially rational activation which is able to model sharp changes.


[1] Elfwing, Stefan, Eiji Uchibe, and Kenji Doya. "Sigmoid-weighted linear units for neural network function approximation in reinforcement learning." Neural Networks 107 (2018): 3-11.

[2] Clevert, Djork-Arné, Thomas Unterthiner, and Sepp Hochreiter. "Fast and accurate deep network learning by exponential linear units (elus)." arXiv preprint arXiv:1511.07289 (2015).

[3] Boullé, Nicolas, Yuji Nakatsukasa, and Alex Townsend. "Rational neural networks." Advances in Neural Information Processing Systems 33 (2020): 14243-14253.

**Summary Of The Paper:**

This paper addresses an important problem of building high-precision regressor for particle physics. The paper designs the sk-DNN, which uses residual layer, and with physics-informed normalization for the data as input. It compares vanilla DNN and with baseline of boosted decision trees. The paper shows that for larger dimensional input (4D, 8D), the sk-DNN shows improved performance and speeds up simulations by a factor of 10^3 – 10^6 over the first-principles computations currently used in Monte Carlo simulations.

**Summary Of The Review:**

In summary, the paper addresses an important problem in particle physics, and achieves good accuracy and significant speedup compared to first-principles computations. The innovation in the ML may be limited, which may be improved with more empirical baselines.

---

> ### Author Response · Authors · 2022-11-18
> **Response for reviewer 3**
>
> We thank the reviewer for the very useful comments that have been made in this review. In fact, we revisited our choice of activation functions and realized that we can further improve the accuracy of our models. We have updated and refined our analysis and the modification are appearing in the updated version of the draft. Below we will leave our comments on some of the concerns raised by the reviewer.
> _____________
> - “On the other hand, in terms of machine learning, the novelty is limited, since the residual connection is a commonly used architecture. The physics-informed normalization is interesting but may be limited in its generality to more general problems. ”
>
> The reviewer is correct about the fact that skip connections are very widely used in several domains of machine learning and they have been around for a while. We do not claim any novelty in proposing its use. However, we are not aware of its use in high-precision applications or in a simple dense neural network. In a way, our proposal of sk-DNN is a very straightforward application of the Highway Networks as we mention in the draft but is an architecture that hasn’t been used or proposed before.
>
> Correlation between functions in a perturbative expansion is not so uncommon since there are often symmetry arguments that connect various terms of a perturbative expansion. We have added an appendix to the paper explaining in greater detail the motivation behind our work and how it is applicable to a much broader spectrum of physics problems.
>
> _______________________
> - “For example, the authors may explore how activation function affects the performance since activation is extremely important in regression tasks. In my experience, for regression tasks, ReLU does not necessarily work well since it essentially models a piecewise linear function. The activation of leakyReLU, SiLU [1], ELU [2], Rational activation [3] are typically very good candidate activation for regression tasks, especially rational activation which is able to model sharp changes.”
>
> We are really grateful to the reviewer for this suggestion. While we had left a comment in the conclusion staging our intent to explore various activation functions in a future work, we decided to go ahead and try this out. We found that the Leaky ReLU function does much better than the ELU, ReLU or the softplus function. We did not try any trainable activation function like the rational activation function suggested by the reviewer but intend to do so in the future. We have updated all our results and find that the DNNs and sk-DNNs definitively outperform BDTs at higher dimensions. We also find that they exceed the accuracy threshold we set with fewer parameters. We can now also show that sk-DNNs require fewer parameters to perform equivalent to a more complex DNN.
> __________________
> In addition to this, we have also made more hyperparameter surveys for the BDTs. We have added a plot on the data size dependence for all architectures and expanded on the physics motivation behind this work. We thank the reviewer once more for the very useful and pertinent feedback.

---

> > ### Comment · Reviewer_XeYb · 2022-11-18
> > **Response**
> >
> > Thanks for making the changes! I appreciate the work the authors do to improve the paper. However, right now I do not find the appendix. Please include it in the pdf for the main paper, after the reference.

---

> > > ### Author Response · Authors · 2022-11-18
> > > **paper in preparation**
> > >
> > > Dear Reviewer,
> > >
> > > We are in the process of finalizing the updated draft. We will upload it by the deadline. Thank you for your patience.
> > >
> > > the authors

---

> > > ### Author Response · Authors · 2022-11-19
> > > **new version uploaded**
> > >
> > > Dear reviewer,
> > >
> > > The new version of the draft is now online. Thank you.
> > >
> > >
> > > the authors

---

> > ### Comment · Reviewer_XeYb · 2022-11-22
> > **Update of the review**
> >
> > I have read the rebuttal of the work. The reviewer mostly addressed my concerns with new experiments.
> >
> > In my opinion, the paper is borderline, with the strength of significance in high-energy physics and the simplicity of the method, and weakness of novelty in terms of machine learning. I'm leaning towards accept considering that this has important application in the high-energy physics and can help our understanding of matter, and is novel in the area of Machine Learning for Science.

---

> > > ### Author Response · Authors · 2022-11-23
> > > **response for reviewer 3 on updated version**
> > >
> > > Dear reviewer,
> > >
> > > We thank you for reviewing the revised version and giving us a positive response. Indeed, the reason why we wanted to submit this paper to ICLR in the applications section is that fast surrogate models for Monte Carlo simulations have a broader field of applicability in the physical and biological sciences beyond just Particle Physics and is a challenge that must be addressed. We hope our example from Particle Physics, with simulated data for real physical systems, will serve as a precursor for more work on this topic.
> > >
> > > best regards,
> > > the authors.

---

> > ### Comment · Reviewer_XeYb · 2022-11-23
> > **Update of the review**
> >
> > I have read the rebuttal of the work. The reviewer mostly addressed my concerns with new experiments.
> >
> > In my opinion, the paper is borderline, with the strength of significance in high-energy physics and the simplicity of the method, and weakness of novelty in terms of machine learning. I'm leaning towards accept considering that this has important application in the high-energy physics and can help our understanding of matter, and is novel in the area of Machine Learning for Science.

---

### Official Review · Reviewer_D2vG · 2022-10-25

**Confidence:** 4
**Correctness:** 4
**Technical Novelty And Significance:** 3
**Empirical Novelty And Significance:** Not applicable
**Recommendation:** 5

**Clarity, Quality, Novelty And Reproducibility:**

The intro and related works is relatively clear, although a bit vague for someone unfamiliar with particle physics expansion.  Is this solving all types of physics problems, or is it looking at a specific decay/scattering/reconstruction/etc. process?  It would perhaps help to put additional details in the appendix to further clarify the functions being expanded/regressed.  Similarly, giving a bit more detail, particularly at the end of section 2 as to how this approach is different would be helpful.  The authors reference they are proposing novel methods for scaling, but novel methods are assumed- can you more specifically state what these are here so the reader has something more tangible in mind as they move into section 3?

4.1 could use additional clarity- it's unclear what these dimensions and functions correspond to physically and whether the values in Table 1 are empirically known, set by the simulator, required by physics, etc.  Is there a choice of the number of dimensions to work in, or is this specified by the given problem (i.e. event to be modeled)?


**Strength And Weaknesses:**

Strengths:
Overall the paper is clear, well motivated, and well written textually (although some clarity around the physics could be provided- see next section).

The authors compare several approaches including BDT, and two NN formulations.


Weaknesses:
 The architectures explored/chosen are not particularly novel.  That's fine if it's not the main focus, but the emphasis seems to suggest that the use of skip connection based architectures is a key insight/contribution.

Additional clarity around the physics would be helpful (see next section).

The authors briefly reference permutations of external processes and over-completeness in 4.1.  Would using a permutation-invariant architecture help to alleviate this problem?

Demonstrating the performance of the different methods as a function of dataset size could be valuable.

How would this extend to other calculations?  The authors have demonstrated this is a good method for this calculation by comparing to the MC based outcome, but what parameters could change to give someone implementing this method that this approach is still valid?  MC methods provide error ranges on the estimates.  How would that work here- what is the error around the physical parameter of interest to be measured and how can you "prove" this?

Only two NN architectures are used.  Would using a more "sophisticated" architecture likely lead to a better result for that approach?

**Summary Of The Paper:**

The authors leverage symmetry principles to design high-precision regressors which enable a dramatic speedup of simulations vs. traditional Monte Carlo approaches.  They demonstrate that BDT performs well in 2D and 4D, but DNN with skip connections are needed for performance in higher (i.e. 8D).

**Summary Of The Review:**

The overall motivation of the paper is clear- to replace time consuming MC methods with faster ML methods.  However, because specific detail around the problem to be solved isn't given, it's not clear why some of the design choices are made.  The impact of dataset size is an important feature that's worth exploring.  While MC methods are slow, they do offer a "know what you're getting" and ability to quantify the uncertainty; it's not clear to me how certain any specific measurement under the proposed framework is without also comparing to MC at some point (i.e. how well does this generalize to different events)?

---

> ### Author Response · Authors · 2022-11-16
> **Response for reviewer 2**
>
> We thank the reviewer for taking a careful look at our work. We would like to address some of the concerns raised.
>
> ___
> - “Weaknesses: The architectures explored/chosen are not particularly novel. That's fine if it's not the main focus, but the emphasis seems to suggest that the use of skip connection based architectures is a key insight/contribution.”
>
> The primary focus of the work is to find the best combination of architecture, training and data strategies that will allow us to build high-precision regressors. An NN architecture with skip connections is just one of the pieces necessary to get this done.
> ___
> - “Additional clarity around the physics would be helpful (see next section).”
> -"The intro and related works is relatively clear, although a bit vague for someone unfamiliar with particle physics expansion. Is this solving all types of physics problems, or is it looking at a specific decay/scattering/reconstruction/etc. process? It would perhaps help to put additional details in the appendix to further clarify the functions being expanded/regressed. Similarly, giving a bit more detail, particularly at the end of section 2 as to how this approach is different would be helpful. "
>
> We are adding a section in the appendix that explains the physics motivations better. This should help in increase the clarity of the physics background of our work.
>
> ___
> - “The authors briefly reference permutations of external processes and over-completeness in 4.1. Would using a permutation-invariant architecture help to alleviate this problem?”
>
> We thank the reviewer for bringing this up. We did consider this at the initial stages of the work. However, for these classes of function and physics processes that we address, the symmetry argument reduces the number of functions that need to be regressed and the symmetry does not need to be built into the neural networks. That being said, we intend to consider this in a future work.
> ___
> - “Demonstrating the performance of the different methods as a function of dataset size could be valuable.”
>
> We showed this for the BDTs in figure 3 right panel. We are adding a plot to show this for the NN architectures.
>
> ____
> - “How would this extend to other calculations? The authors have demonstrated this is a good method for this calculation by comparing to the MC based outcome, but what parameters could change to give someone implementing this method that this approach is still valid?”
>
> The methods that we explore are general in the sense that they can be used to regress any function that is expensive to compute in an MC but a large sample can be obtained for once to train an ML model that will later speed up the MC. Of course, a certain degree of tuning is required for some cases. However, with what we have, we can address a lot of particle physics processes or any other processes that are cast as perturbative expansion with the higher-order terms taking a lot more time to compute, which is usually the case in perturbation theory.
>
> ____
> - “MC methods provide error ranges on the estimates. How would that work here- what is the error around the physical parameter of interest to be measured and how can you "prove" this?”
>
> A very good point. The MC samples that we generate actually come with errors. We are assuming that a very large sample can be generated from the MC (~10M in our case) with which the ML algorithm is trained. The ML algorithm itself will predict with a certain error the distribution of which can be computed from a test sample as we show in figure 4. This can be used to estimate the “systematic” error due to using a regressor and it is this systematic error that we want to reduce by making the regressor more accurate. Then the error can be cast as an MC error (known from generating the sample) + systematic error from the regressor.
>
> ____
> - “Only two NN architectures are used. Would using a more "sophisticated" architecture likely lead to a better result for that approach?”
>
> We are in the process of trying some variations to the architecture but have not come across anything that is significantly better. So we do not have an answer to this question right now but we will keep searching.
>
> “While MC methods are slow, they do offer a "know what you're getting" and ability to quantify the uncertainty; it's not clear to me how certain any specific measurement under the proposed framework is without also comparing to MC at some point (i.e. how well does this generalize to different events)?”
>
> There is certainly a problem with error quantification in regions not explored in the sample used to train the NN. However, if this work we generate very large datasets that completely cover the domain of the function and are hence quite representative. The goal is to generate such large samples just once and then find the best regressor possible instead of generating equivalently sized or much larger samples several thousand times as is normal in the lifecycle of a particle physics analysis.

---

### Official Review · Reviewer_P5Fn · 2022-10-25

**Confidence:** 5
**Correctness:** 1
**Technical Novelty And Significance:** 2
**Empirical Novelty And Significance:** 1
**Recommendation:** 3

**Clarity, Quality, Novelty And Reproducibility:**

The paper as a whole is relatively easy to follow. However, at places it is not easy to understand the motivation or the introduced approach. The quality of experiments are very poor and I strongly believe that even though the results might be reproducible, it is not supporting the claims in the paper. I have some serious reservations when it comes to novelty.

**Strength And Weaknesses:**

Strengths:
1-	Utilizing domain knowledge in ML process
2-	Might be a problem of significance importance.

Weakness:
1-	Unsupported claims and decisions.
2-	Lack of novelty
3-	Poor experimental study


**Summary Of The Paper:**

The authors aim to do regression for particle physics that has high precision over the domain of the function. The authors claim that the standard data normalization and regression models are not precise for this application. The authors introduce a normalization method and a fully connected NN with skip connection and claim that their approach is better than the standard treatment for this problem.

**Summary Of The Review:**

The authors introduce three properties for a desirable regresson in the beginning of Page 2. However, they do not explain why these specific properties are needed. More particularly, it is not clear what practical implication the high precision property that says errors should be less than 1% over more than 90% of the domain of function has. Where do the numbers 1% and 90% come from? Also, why lightweight property is important?

I could not make sense of the normalization introduced in the paper. I am not even sure what Figure 1 is showing. The authors also failed to show that the new introduced normalized helps improving the performance in practice.

Why not directly optimizing the relative error? Couldn’t that address the problem with normalization?

From Figure 3, it seems that a lower learning rate than 0.01 is needed as the lower values generates better results. Why aren't values lower than 0.01 tested? Similarly from Figure 4 for BDT, it seems that a lower value for max-depth is preferred. Note that for gradient boosting in general, weaker base models such as a decision stump (decision tree with one node) is preferred. I wonder why values lower than 10 is not tested. They later use the max depth of 50 for the baseline model of BDT in Table 2. This is definitely a wrong depth value for boosting models. In the end of the day, all these hyperparameters need to be optimized using the validation set.

I am a bit confused how the data are split. In Section 3.1., the authors say they used 60% for training and 40% for test. Later on the bottom of Page 6, they say they used 48% for training, 32% for validation and 20% for test. Also, I wonder how the authors came up with these specific non-standard splits. Yet again, they change their data split for the baselines in Section “Baselines” in Page 7 to 80% training and 20% test. So much inconsistencies everywhere.

I am not sure why the authors did not test the same values for depth and width for both DNN and sk-DNN. It is hard to compare the results. In any case, it seems that regular DNN performs better than sk-DNN based on the results from Figure 4.

---

> ### Author Response · Authors · 2022-11-16
> **Response to reviewer 1 (2 of 2)**
>
> __(continued from previous post)__
>
> - “ it seems that a lower value for max-depth is preferred. Note that for gradient boosting in general, weaker base models such as a decision stump (decision tree with one node) is preferred. I wonder why values lower than 10 is not tested. They later use the max depth of 50 for the baseline model of BDT in Table 2. This is definitely a wrong depth value for boosting models. In the end of the day, all these hyperparameters need to be optimized using the validation set.”
>
> We are afraid the reviewer is mistaken. Figure 4 clearly shows that a max_depth of 50 is preferred at 2D and 4D and 20 is preferred at 8D since the density plot is much more peaked for higher max_depth. Table 2 also shows that BDTs with larger depths have much lower errors. Please, take a careful look at this.
>
> The notion that shallower trees work better comes from the point of view of generalization when a lot of noise is present in the data. However, with physics MC simulation, the noise in the data is usually very low and the datasets are very large (10M in our case). In these cases, BDTs with larger depth far outperform BDTs with smaller depth. We will add a plot with tests we did for lower depths as low as 1 and we clearly see a very sharp reduction in performance.
> ____________________________
> - “I am a bit confused how the data are split. In Section 3.1., the authors say they used 60% for training and 40% for test. Later on the bottom of Page 6, they say they used 48% for training, 32% for validation and 20% for test. Also, I wonder how the authors came up with these specific non-standard splits. Yet again, they change their data split for the baselines in Section “Baselines” in Page 7 to 80% training and 20% test. So much inconsistencies everywhere.”
>
> We have very large datasets (10M for training) that covers the whole domain of the functions and hence we can afford to have large validation sets. In section 3 we refer to the data we use for training the models and the hyperparameter tuning. Of this set 60% is used for training and 40% is used for the validation of the models. Yes, this is non-traditional and we specify that this is needed to increase the accuracy of the models and it is feasible with the large datasets we have. Instead for the baseline, we use an 80%-20% split since we want to show the results with a more traditional split. Furthermore, we set aside some data just for testing the model after they have been trained. This set is separate from the training or the validation set. We call it the test set. So, from the whole data sample (100%), 20% is test set, and 80% is used for training + validation. Of the 80%, 60% is used for training and 40% for validation, which make is 48% of the whole dataset for training and 32% of the whole dataset for validation. 48% (training) + 32% (validation) + 20% (test) = 100% (whole dataset). We will clarify this further in the next version.
> _________________________
> - “I am not sure why the authors did not test the same values for depth and width for both DNN and sk-DNN. It is hard to compare the results. In any case, it seems that regular DNN performs better than sk-DNN based on the results from Figure 4.”
>
> We mention in the text that we would like to test a similar number of parameters for the DNN and sk-DNN since we want to test NNs of similar complexity. For the DNN, the depth refers to the number of layers and for the sk-DNN the depth refers to the number of blocks. In Table 2 and in figure 4 we specify the number of parameters that each model has and we compare models that have a similar number of parameters.
>
> Figure 4 shows that sk-DNNs outperform DNNs of similar complexity . The higher the peak and the narrower the width of the error distribution the smaller the errors are over the entire domain. These are density plots of the errors. We would like to remind the reviewer that $\delta$ is the relative error and smaller relative errors imply better models.
> ___________________
> We would like to thank the reviewer for the time and effort spent. However, we feel that the reviewer should take a careful look at figure 4 and table 2 to see that they show what we claim as the conclusions of this work. Please keep in mind the plots are of the error distribution. We have taken several of the reviewer's comments into consideration and are adding appendices and plots to further clarify our point.

---

> ### Author Response · Authors · 2022-11-16
> **Response to reviewer 1 (1 of 2)**
>
> We thank the reviewer for the comments. We would like to respond to some points the reviewer has raised. We quote the reviewer below and respond to the concerns point by point.
>
> - “The quality of experiments are very poor and I strongly believe that even though the results might be reproducible, it is not supporting the claims in the paper.”
>
> We would like to point out that the results we claim are shown directly in the plots and tables we present for the results. We elaborate on this below.
> ______________
> - “The authors introduce three properties for a desirable regression in the beginning of Page 2. However, they do not explain why these specific properties are needed. More particularly, it is not clear what practical implication the high precision property that says errors should be less than 1% over more than 90% of the domain of function has. Where do the numbers 1% and 90% come from? Also, why lightweight property is important?”
>
> Precision requirements: The functions that we regress are used in Monte Carlo generators to simulate particle physics processes at colliders like CERN. In the future, these colliders will achieve a certain precision for the experimental observables. and this has been estimated through decades of research in particle physics. The experimental precision achievable determines the precision requirements for the theoretical estimates made using Monte Carlo generators. We do not derive these numbers in this paper, since it is not in the scope of the work, We use this precision as a benchmark to prove that it can be done.
>
> Lightweight: The regressors that we are making not only need to be fast they also need to be portable as they will be part of a Monte Carlo simulation code. For example, the BDT that is created for the 8D datasets can be several hundred MB to several GB in size when stored as a compressed model. This is not possible to ship as a code since several of these regressors will be necessary for a single MC process. On the other hand, the DNN weights are a few KB making them very portable. We had to take this into consideration since we are planning on a real-world application.
>
> We are adding an appendix where we will explain the physics case better.
>
> ____________________
> - “I could not make sense of the normalization introduced in the paper. I am not even sure what Figure 1 is showing. The authors also failed to show that the new introduced normalized helps improving the performance in practice.”
>
> Figure 1 shows that $f_{00}$ and $f_{02}$ are both very peaked as their values span several orders of magnitude and are also very highly correlated ($\rho$ ~ 0.9) as can be read from the axes and as we also mention in the text. Once $f_{02}$ is normalized by $f_{00}$ the resulting function, $f$, is not as peaked and spans only one order of magnitude as shown by the color bar on the right of each panel. The improvement in performance of each architecture due to the normalization can be seen in Figure 4 where the “baseline” (solid blue density plot) is without using the normalization and is much broader than the other density curves for which the normalization is used.
>
> _______________________
> - “Why not directly optimizing the relative error? Couldn’t that address the problem with normalization?”
>
> We tried this and this does not help. The reason we optimize with MSE is because it is known that MSE heaviliy penalizes outliers and we do not want to have large outliers. The functions we regress are kernels of integrals and large outliers can throw off the integral in an unexpected manner. Hence we optimize with MSE but show our results in terms of relative error. The normalization problem affects all architectures and cannot be removed by changing the optimization metric.
>
> _______________________
> - “From Figure 3, it seems that a lower learning rate than 0.01 is needed as the lower values generates better results.”
>
> Figure 3 shows that lowering the learning rate below 0.01 does not help since the curve plateaus. Further lowering the learning rate not only adds no improvement, it also increases the number of trees in the BDT as the learning rate and ensemble size are correlated when early stopping is used to stop the growth. We will have a plot showing this explicitly for lower learning rates in the upcoming revision.
> ____________________
> _continued..._

---

### Comment · Area_Chair_KyJH · 2022-11-15
**Please engage before the author-reviewer discussion closes**

Dear authors and reviewers,

The first phase of the discussion period is about to close on November 18.

For authors, please make sure to submit your rebuttal by the deadline. Leave some time for the reviewers to read it and respond while you are still allowed to further engage with them. Interactions between authors and reviewers are very important for the quality of the review process, so please make sure to engage.

For reviewers, please try to acknowledge and respond to the authors' rebuttal while the discussion period is still open for them to further interact with you.

Thank you for your participation in the review process!

Best,
The AC

---

> ### Author Response · Authors · 2022-11-15
> **Rebuttal in preparation**
>
> Dear Area Chair,
>
> Thank you for the reminder. We are preparing rebuttals for the reviewer's comments and also making significant modifications to the draft to address their concerns. We will be posting our work soon.
>
> best regards,
> the authors

---

> ### Author Response · Authors · 2022-11-19
> **Summary of Changes**
>
> The following are the changes we made to the draft in response to the reviewer’s comments.
>
> - We have redone all the analyses for the DNN and the sk-DNN using Leaky ReLU instead of ReLU in accordance with the suggestions of the third reviewer. We now see that the DNNs perform better than the BDTs at 4D and significantly better at 8D. We also see that the DNN and sk-DNN exceed the precision requirements we set for 2D, 4D and 8D. This is a significant improvement over what we had got before with ReLU
>
> _____________
> - We fixed an issue with the 8D analysis pipeline for BDTs. Hence the corresponding plots and numbers are different. We also see that a BDT of depth 20 performs better than a BDT of depth 50.
>
> _______________
> - We added 3 appendices. The first one better explains the physics motivation and logic behind the parameters we set for this analysis. It also explains why these regressors are more generally applicable in several other areas of physics. The second one elucidates the hyperparameter decisions made for the BDTs. The third one shows a comparison plot with various activation functions for the neural networks.
> ________________
> - We have changed figure 3 to include a data volume requirement comparison between the BDT, sk-DNN and DNN for 2D, 4D and 8D data in response to the third reviewer.
> ___________________
> - We have made several changes to the test to make it clearer to understand our objectives and results in response to several of the reviewers.
> ____________
> - A typo in eq. 3 has been corrected. The mathematical notations have been made more uniform following the suggestion of reviewer 1.
> _______________
>
> - all figures and tables have been updated with the results we got due to the changes suggested by the reviewers.
>
> ____________
> We thank all the reviewers for their time and effort and thank the chairs for organizing and managing this. We hope our modification adds further clarity to the objective we had in mind for this analysis.

---

### Author Response · Authors · 2022-11-24
**Novelty of our work vis-à-vis Machine Learning for Science**

One popular use of Monte Carlo (MC) simulators is to create density estimates of distributions of variables that cannot be computed analytically or when these computations are intractable due to their complexity. To perform these simulations certain set of functions have to be computed numerically a very large number of times as we mention in the abstract and around eq. 1 in the introduction. Most of the works that we cite, try to address this problem by proposing generative models (GANs, VAEs) to estimate the final distribution of the variables being estimated, effectively removing the need for an MC simulation having completely replaced it with a generative model. However, these attempts have failed to reach the desired level of accuracy (compared to MC simulations), especially at higher dimensions.

Our method takes a different approach. Instead of trying to replace the MC simulation, we look into the functions that create a bottleneck for the MC simulation. We build regressors for those functions which is a more tractable task and helps the MC simulation do what it does best, generate density estimates from the regressed functions. So instead of replacing the MC simulation with a generator, we propose building in high-precision regressors within the MC simulator so that large samples can be generated much faster. We believe this is a shift in the way the problem is being addressed using machine learning and we felt it apt to submit it to the Machine Learning for Science track. We hope the reviewers and the area chairs have a more favourable view of the novelty we present and we are grateful for their comments which have helped us improve our work and present it more clearly.

---

### Decision · Program_Chairs · 2023-01-20

**Decision:**

Reject

**Justification For Why Not Higher Score:**

Lack of novelty and significance for ML.

**Justification For Why Not Lower Score:**

N/A

**Metareview: Summary, Strengths And Weaknesses:**

Reviewers either strongly recommend rejecting the paper (3-1) or are unsure and place the paper in the borderline category (5-6). This paper explores the acceleration of particle physics simulations by building surrogate regressors replacing parts (here, helicity amplitudes) of the MC simulation.  The major concern raised by the reviewers is a clear lack of novelty and significance for the ML community.  Indeed, the paper evaluates traditional regression methods (decision trees, MLPs, MLPs with skip connections) and reaches expected results: DTs perform best in low-dimensional regimes while DNNs perform better in high-dimensional regimes.  The significance in particle physics is also unclear, as the authors do not compare their results to any other previous attempts at surrogate modeling. While the experimental validation is good and optimizing this specific part of the physics simulation is certainly worth implementing, it is difficult to assess whether the approach and conclusions of the paper are generalizable to other problems in ML and science.